# A Phase Generation Shifting Algorithm for Prosumer Surplus Management in Microgrids Using Inverter Automated Control

**Ovidiu Ivanov \*** , **Bogdan-Constantin Neagu** , **Mihai Gavrilas and Gheorghe Grigoras**

Power Engineering Department, Gheorghe Asachi Technical University of Iasi, 700050 Iasi, Romania;
bogdan.neagu@tuiasi.ro (B.-C.N.); mgavril@tuiasi.ro (M.G.); ggrigor@tuiasi.ro (G.G.)
\* Correspondence: ovidiuivanov@tuiasi.ro

**Abstract:** Four-wire low-voltage microgrids supply one-phase consumers with electricity, responding to a continuously changing demand. For addressing climate change concerns, national governments have implemented incentive schemes for residential consumers, encouraging the installation of home PV panels for covering self-consumption needs. In the absence of adequate storage capacities, the surplus is sold back by these entities, called prosumers, to the grid operator or, in local markets, to other consumers. While these initiatives encourage the proliferation of green energy resources, and ample research is dedicated to local market designs for prosumer–consumer trading, the main concern of distribution network operators is the influence of power flows generated by prosumers' surplus injection on the operating states of microgrids. The change in power flow amount and direction can greatly influence the economic and technical operating conditions of radial grids. This paper proposes a metaheuristic algorithm for prosumer surplus management that optimizes the power surplus injections using the automated control of three-phase inverters, with the aim of reducing the active power losses over a typical day of operation. A case study was performed on two real distribution networks with distinct layouts and load profiles, and the algorithm resulted efficient in both scenarios. By optimally distributing the prosumer generation surplus on the three phases of the network, significant loss reductions were obtained, with the best results when the generated power was injected in an unbalanced, three-phase flow.

**Keywords:** electricity distribution; microgrids; prosumers; phase generation management; metaheuristic optimization

## 1. Introduction

Residential consumers that use photovoltaic (PV) panels for local generation can sell their generation surplus back to the grid, using one of two main trading options: reselling back to the network operator at regulated tariffs [1] or using local markets for trading to local consumers [2]. The first method is mostly benefitting the suppliers, because the trading prices are lower than in the deregulated wholesale electricity market. In Romania, for instance, the reference price uses the average day-ahead market price over the last year [3]. The local trading model mainly benefits consumers, who can buy electricity at lower prices than those offered by traditional suppliers, but also prosumers, who can obtain higher electricity purchase prices from consumers.

From a technical standpoint, the prosumers' microgrids are able to gradually gain independence from the grid and monopoly utilities. However, this advantage comes with a cost to the utilities, because, as long as the microgrid is still connected to the distribution network, it can influence its operating conditions. When the supply is provided exclusively by large-scale remote generation (coal, hydro, nuclear), power flows are unidirectional, from the source to the consumer. When distributed generation is present in the network, the local generation sources can bring significant changes to the power flows, with new challenges for the security and quality of the supply. Two main scenarios can occur:

- The local generation is lower than the total consumption in the network. In this case, power flows are reduced in the microgrid, the local demand being satisfied by the closest proximity. However, losses can still increase in areas with an important surplus due to supplemental power flows.
- The local generation exceeds the consumption. In this case, the power flows are reversed, with high changes in the operation conditions of the network, affecting both power losses and quality of supply. In this case, the network is operated in conditions for which it was not designed for.

Even without the presence of prosumers, four-wire low-voltage (LV) networks are already operated with unbalanced phase loads, because they supply mainly one-phase consumers with variable hourly demand. The local generation by the prosumers has the potential of increasing the unbalance, because of the inherent unpredictable pattern of renewable generation [4]. These changes can lead to increased energy losses and undesired bus voltage variations. The simplest approach for reducing losses caused by prosumers is to minimize their interaction with the grid, by optimally using the local generation [5]. Since this is difficult to achieve in most situations, the network operator must find other ways to manage the changes. Studies have been performed that aim to determine the optimal prosumer-to-consumer ratio related to the size of the microgrid. The results of [6] show that prosumer-to-consumer ratios in the range of 40–60% have the best performance, with improved self-consumption ratios and self-sufficiency ratios for microgrids due to aggregation effects. Other studies imply that the type of trading (centralized or peer-to-peer) negligibly affect power losses [7]. Thus, the utilities can resort to investing in new equipment or to using optimization techniques for managing consumption and local generation.

Traditional methods consider the optimal reconfiguration and reactive power control, such as in [8]. Some approaches consider the optimal use of energy storage, which can be individual [9] or shared [10,11]. In a study [12], the presence of storage was shown to improve the network state by using it to minimize the generation/demand unbalance, rather than the cost of electricity. The supplemental storage resource provided by electrical vehicles was proposed for optimizing the operating state of the network in another work [13]. In [14], the prosumers were regarded as "provisional microgrids" and used to supply electricity for sustaining the operation of microgrids in islanded operation scenarios. Local markets for ancillary services were also proposed [15]. A more radical change was suggested in [16], by replacing traditional LV networks with DC microgrids. To account for the losses caused in the grid, a study [17] proposed that they should be mitigated by energy transactions in the market, while the formation of "coalitions of microgrids" was envisioned in [18]. A coordinated control scheme in which the network operator uses price signals to induce prosumers' behaviors was proposed in [19]. A similar price mechanism was also used in another study [20].

The inverters of PV systems transform DC voltage into AC voltage, used by the prosumers to feed the surplus in the unbalanced LV microgrid. However, they can also be used for improving the operating conditions of the network. In [21], the output of single-phase inverters was regulated by the means of an electronic network PQ controller to improve voltage and current balance on the phases. In another paper [22], they were used as reactive power compensation devices for voltage quality improvement. The phase shifting of the inverter output voltage with respect to the grid voltage, in order to control the power factor with a minimum number of devices, was studied [23]. The optimal harmonic filtering in inverters was also researched [24]. In [25], the focus was on data sharing between pieces of equipment for achieving network-wide control. In [26], the control was extended on two voltage levels, MV and LV. A novel model to define and co-optimize the deliverable energy flexibility and frequency regulation capacity of power distribution systems was developed [27] by considering a queuing system, energy storage (ES) devices, and distributed solar resources with controllable inverters.

The inverters built in three-phase technology are becoming common in PV systems. They are used in applications such as fault management [28], current and voltage regulation [29], and maximum power point tracking [30].

From the computational standpoint, the current research shows that prosumer management is difficult to solve accurately. A study [31] concluded that prosumer scheduling in microgrids is an NP-hard problem, to which approximate solutions can be found. Thus, if the prosumer management problem (PSM) can be formulated as an optimization problem, the computation-intensive classical optimization algorithms can be replaced with other methods with a minimal tradeoff in accuracy.

Taking into consideration all of the above, this paper focuses on optimizing the power injections of prosumers connected in three-phase low-voltage distribution networks, aiming for the minimization of active power losses over a typical day. For each prosumer, the algorithm needs to determine the amount of power supplied on each phase, using for this purpose a metaheuristic optimization technique, the Particle Swarm Optimization (PSO).

The main contributions of this paper are:

- the design of the PSO-based optimization algorithm for prosumer surplus management in LV microgrids;
- the use of PV prosumer inverters to regulate three-phase power flows to improve the operation state of the microgrid;
- a comparative case study, using two real distribution networks in Romania, with distinctive geographical layout, size, and consumption characteristics.

The rest of the paper presents, in Section 2, the formulation and assumptions used to solve the prosumer surplus management. In Section 3, the adaptation of the PSO algorithm to the PSM problem is described. Section 4 provides the results of the case study. Conclusions and discussions can be found at the end.

## 2. The Formulation of the PSM Problem

Power loss minimization is pursued as a mean of reducing the cost of operation in classic distribution networks without distributed generation. The utility applies measures such as balancing the consumption on the phases of the network, by optimally distributing the single-phase loads on the three phases, or voltage regulation in the MV/LV substation, by modifying the transformer tap settings to compensate for voltage drops on the feeders.

However, the positive effects of these measures may be affected by the intermittent presence of power injections from the prosumers, which can change the balance of power flows on the three-phase network. This disturbance can have as consequences the increase of the level of losses or the worsening of the phase voltages. These effects are to some extent correlated, with high values of losses being obtained in networks with unbalanced consumption on the three phases, which also leads to an unbalanced bus voltage profile.

The prosumers can inject their surplus power into the network in two typical scenarios:

- Unbalanced one-phase, on the connection phase where the demand is located, in which case the overlapping power injection can contribute to the accentuation of the load imbalance on phases (Figure 1a);
- Symmetrical three-phase, in which case the influence of the prosumer on the phase load balance is negligible (Figure 1b).

This paper considers another assumption, in which the prosumers equipped with controllable three-phase inverters, as part of the smart grid infrastructure, are able to inject the excess generation in an unbalanced manner on the three phases (Figure 1c). Thus, the prosumer can achieve its goal of selling its surplus back to the supplier or to other consumers in the local market, contributing at the same time to the optimization of the operating conditions in the network through the minimization of power and energy losses.

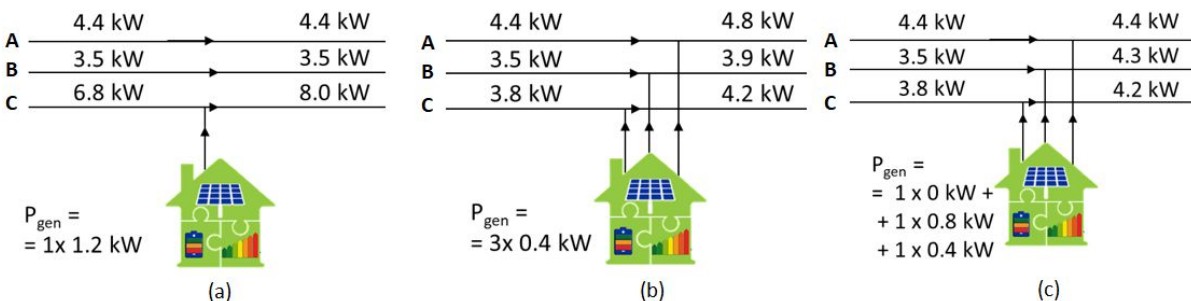

**Figure 1.** The approach used by the PSM algorithm for loss minimization: (**a**) unbalanced one-phase injection, (**b**) three-phase balanced injection, (**c**) unbalanced three-phase injection.

The aim of the algorithm is to find the amount of power surplus generated by each prosumer on each phase, so that the sum of power losses on all the branches of the radial network is minimized:

$$\Delta P = \sum_{h=1}^{hmax} \sum_{br=1}^{NB} \Delta P_{h,br} = min \tag{1}$$

where *NB* is the number of branches in the microgrid; *hmax* is the number of time intervals used in the analysis, and $\Delta P_{h,br}$ indicates the power losses on branch *br* at hour *h*, computed with:

$$\Delta P_{h,br} = K_{br} \cdot R_{ac,br} \cdot I_{h,br}^2 \tag{2}$$

In Equation (2), $R_{ac,br}$ is the phase resistance of the branch *br*, while $K_{br}$ is the coefficient used in Romanian standards to account for the losses on the neutral wire [32], as in Equation (3), and $I_{h,br}$ is the branch current flow, computed with Equation (4):

$$K_{br} = CUF_{br} \cdot \left(1 + 1.5 \cdot \frac{R_{n,br}}{R_{a,br}}\right) - 1.5 \cdot \frac{R_{n,br}}{R_{a,br}} \tag{3}$$

$$[I_{h,br}] = -inv(A) \cdot [I_{h,bus}] \tag{4}$$

Equation (3) uses the current unbalance factor *CUF* from [32], computed for each branch and time interval, where the time index *h* is omitted for simplicity:

$$CUF_{br} = \frac{1}{3} \cdot \left[ \left(\frac{I_{br,a}}{I_{br,abc}^{avg}}\right)^2 + \left(\frac{I_{br,b}}{I_{br,abc}^{avg}}\right)^2 + \left(\frac{I_{br,c}}{I_{br,abc}^{avg}}\right)^2 \right] \tag{5}$$

In Equations (3)–(5), the following notations were used: $R_{n,br}$—resistance of the neutral wire of branch *br*, *A*—branch–node connectivity matrix, $[I_{h,br}]$, $[I_{h,bus}]$— branch and bus currents vector for hour *h*; $I_{br,a}$, $I_{br,b}$, $I_{br,c}$, $I_{br,abc}^{avg}$—phase and average currents on branch *br*. The algorithm formulated in Equations (2)–(5) is applied on each phase, with the currents $I_{h,bus}$ determined using the phase consumptions originating from the network, updated with the contribution of the unbalanced prosumer generation.

## 3. Adaptation of the PSO Algorithm for Prosumer Surplus Phase Shifting

Population-based metaheuristic algorithms are used for optimization problems where approximations of the optimal solutions can be obtained with simple mathematical models and a reasonable calculation time [33]. One of the best-known metaheuristics is Particle Swarm Optimization [34], inspired by the natural movement of large groups. The algorithm works on the principle of changing the position of each individual in the search space by pulling it simultaneously in two directions: towards its best known position and towards the best position ever achieved by the swarm. The closeness to the optimal solution is measured by a fitness function for each individual or particle, and the optimal solution is encoded in the position of the swarm leader, in the last iteration. Figure 2 depicts the basic diagram of the PSO algorithm.

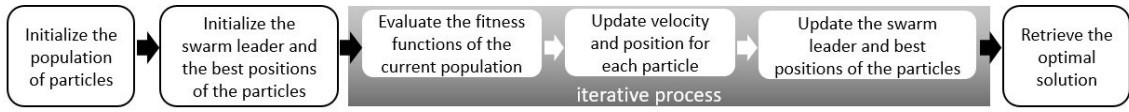

**Figure 2.** Basic flowchart of the PSO.

Its simplicity and versatility have made the PSO one of the most used metaheuristic algorithms in optimization problems [35]. The adaptation for the PSM problem keeps the flowchart from Figure 2, the changes being made at the level of the way in which the initial population is generated and the fitness function is computed. The structure of a particle encodes the percentage of each prosumer surplus generated on each phase, using the model in Figure 3.

| PS1, a | PS1, b | PS1, c | PS2, a | PS2, b | PS2, c | Psi, a | Psi, b | Psi, c | PSn, a | PSn, b | PSn,c |
|--------|--------|--------|--------|--------|--------|--------|--------|--------|--------|--------|-------|
| 44.10 | 49.03 | 6.87 | 55.58 | 38.48 | 5.94 | ... | ... | ... | 46.10 | 7.53 | 46.37 |
| sum = 100 | | | sum = 100 | | | sum = 100 | | | sum = 100 | | |

**Figure 3.** Structure of a particle for the PSM problem.

For each prosumer with generation surplus, the particle contains three values, each for one phase, totaling a sum of 100. In the velocity update step, each particle must be validated to follow this rule, so that the entire surplus generation will be injected in the grid.

The fitness function computes the power losses for each particle according to the procedure presented in Section 2. The particle with the minimal value of the losses obtained at the end of the iterative process is considered as the optimal solution discovered for the microgrid. A time interval of 24 h is used in the analysis, and the consumer load profiles and prosumer generation are measured from the real network using the existing Smart Metering equipment.

## 4. Case Study

The PSM algorithm was implemented on two real LV distribution networks in Romania, to which prosumers with house PV panels are connected, having local generation profiles recorded in the 06:00–18:00 h interval. These networks were chosen to represent two different prosumer operation scenarios. The first one, denoted in the following as R28, has a simple structure, specific for microgrids, with two radial feeders and 28 buses. It supplies a number of 28 residences, with one consumer for each pole. In this network, eight prosumers are present, with hourly generation ranging from 0.429 kW to 5.825 kW. This network is located in the center-east side of Romania, in an area with high PV generation potential, and supplies a suburb of newly built houses. Its one-line diagram is drawn in Figure 4, and summary data are provided in Table 1. The second distribution network used in the study, R121, has a highly branched structure, with multiple secondary feeders and usually more than one consumer connected at the poles, as it can be seen in Figure 5. This network has 121 buses and supplies 113 consumers (Figure 5, Table 2). It contains eight prosumers, with generation ranging from 0.083 kW to 1.942 kW, and is located in the northern extremity of the country, a region with lower PV generation potential and lower economic development (thus reduced household consumption). The buses where prosumers are present are depicted in both networks with inverted colors (black fill, white text).

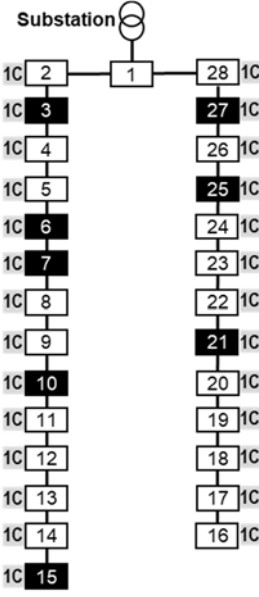

**Figure 4.** Network R28.

**Table 1.** Summary data for network R28.

| Buses | 28 |
|:---:|:---:|
| Consumers | 27 |
| Prosumers | 8 |
| Total load | 835.55/464.25 kW |
| Total prosumer generation | 366.00 kW |
| Total prosumer surplus | 167.97 kW |
| Network type | Overhead, stranded |
| Total/ main feeder length | 1120/600 m |

The networks have distinct demand and generation patterns, and the measured loads and generation are typical for a summer day. Tables 1 and 2 present the aggregate demand in the 24 h interval and in the 06:00–18:00 h interval, the prosumer generation, and the prosumer surplus. The network R28 has higher loads and higher prosumer generation. The consumption in the hourly interval 06:00–18:00 exceeds the PV generation. In the network R121, the consumer demand is much lower, and, in the PV generation interval, total generation exceeds the aggregated demand of all the consumers, a scenario that favors the presence of reversed power flows.

The PSM algorithm was applied to minimize the power and energy losses in two scenarios: shifting the entire surplus of a prosumer on a different phase, with a new static connection, minimal investment, and no requirement for smart grid continuous control, or using a three-phase inverter to distribute the surplus on the three phases.

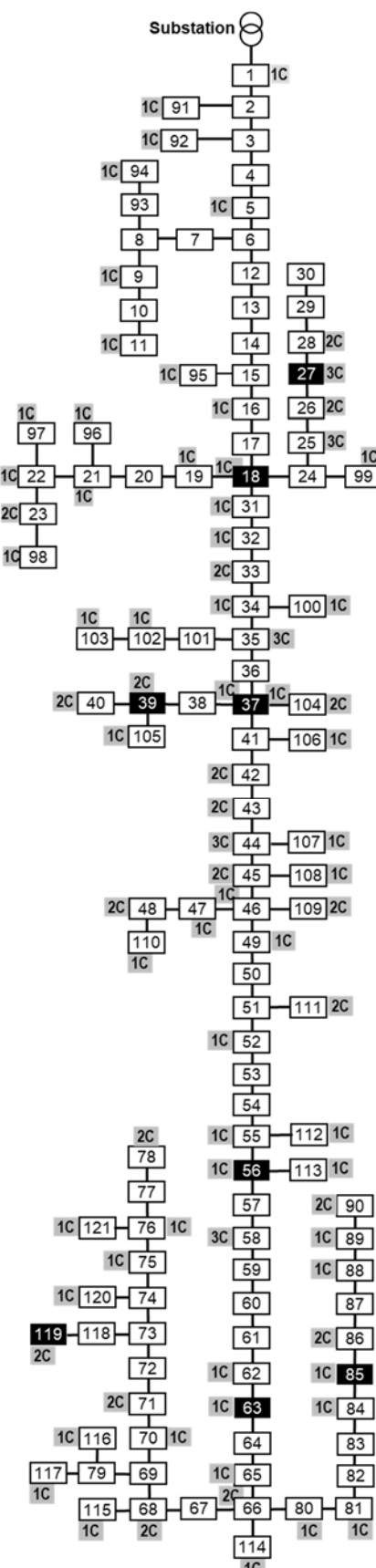

**Figure 5.** Network R121.

**Table 2.** Summary data for network R121.

| Buses | 121 |
| --- | --- |
| Consumers | 113 |
| Prosumers | 8 |
| Total load | 219.85/76.01 kW |
| Total prosumer generation | 122.00 kW |
| Total prosumer surplus | 75.38 kW |
| Network type | Overhead, classic |
| Total/ main feeder length | 4840/2240 m |

*4.1. Results for Network R28*

This network has high consumption, high prosumer generation, and high prosumer surplus that does not exceed the local demand. The demand, generation surplus, and actual bus load (resulting from the aggregation of the demand and the generation profiles using the initial prosumer phase allocation) are indicated for each phase in Figures 6–9.

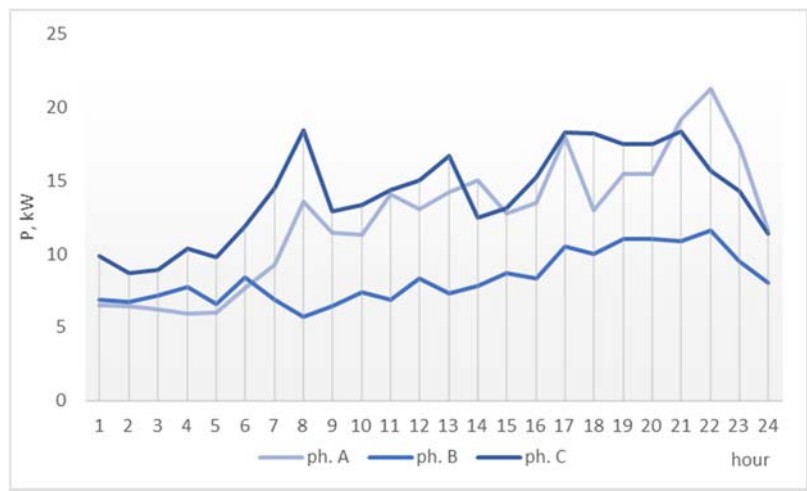

**Figure 6.** Demand in network R28.

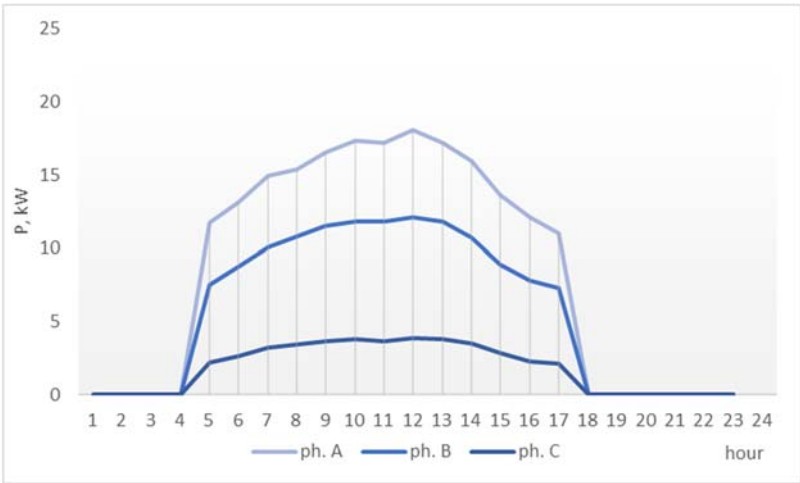

**Figure 7.** Generation in network R28.

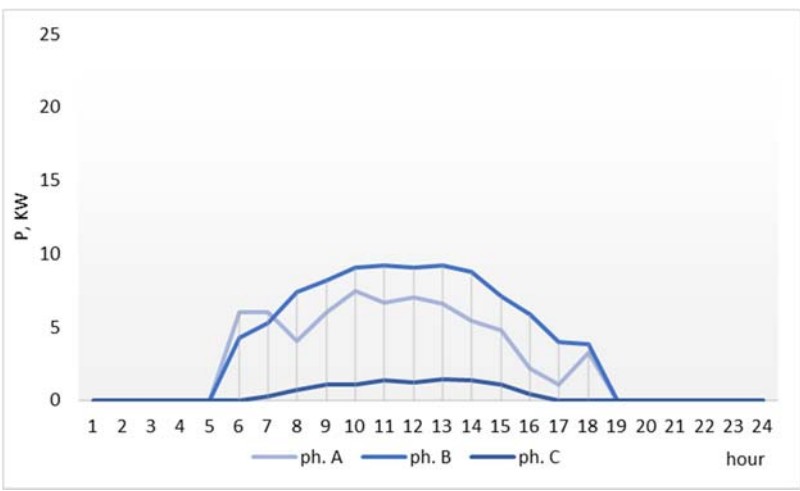

**Figure 8.** Prosumer surplus in network R28.

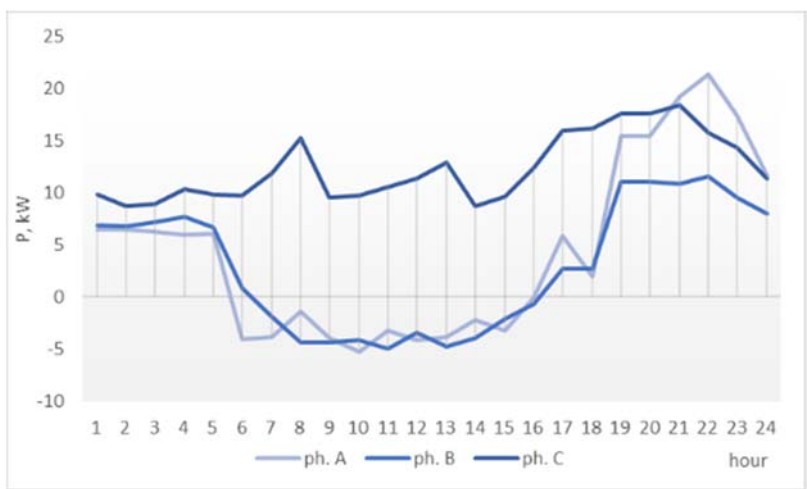

**Figure 9.** Actual bus demand in network R28 (aggregated demand and generation).

The graphs show that, while a part of the prosumer generation is used for covering the self-consumption of the prosumers, a significant surplus exists, that contributes to the increase the phase load unbalance in the network (Figures 6 and 9). Figure 9 shows that on phases A and B, the existing surplus exceeds the demand. If these prosumers are connected to the network using three-phase inverters, the smart grid communication infrastructure could be used by the network operator to control the phase injections so to reduce the active power and energy losses by balancing the phase load.

The new phase distribution of the prosumer surplus, in the two optimization scenarios, is presented in Table 3. The results show that, indeed, both methods of shifting the surplus had a positive effect on improving the reduction of power losses by phase load balancing. Using the initial phase connection of the prosumers, given in Table 4, the total active power loss in network R28 was 8.86 kW. By shifting the entire surplus of the prosumers on a single phase, the losses were reduced to less than half, at 3.82 kW. However, if the proposed method of distributing the surplus on more than one phase is used, the power losses can reach a minimum of 2.40 kW, which represents a significant improvement and cost reduction for the network operator over a larger time interval of months or years.

**Table 3.** Surplus redistribution solutions for network R28, in percent.

| Prosumer | P3 | | | P6 | | | P7 | | | P10 | | | P15 | | | P21 | | | P25 | | | P27 | | |
|---|---|---|---|---|---|---|---|---|---|---|---|---|---|---|---|---|---|---|---|---|---|---|---|---|
| Phase | A | B | C | A | B | C | A | B | C | A | B | C | A | B | C | A | B | C | A | B | C | A | B | C |
| Initial | 0 | 100 | 0 | 100 | 0 | 0 | 0 | 0 | 100 | 100 | 0 | 0 | 100 | 0 | 0 | 100 | 0 | 0 | 0 | 100 | 0 | 0 | 100 | 0 |
| Opt, 1PH | 0 | 0 | 100 | 100 | 0 | 0 | 0 | 0 | 100 | 0 | 0 | 100 | 100 | 0 | 0 | 0 | 0 | 100 | 0 | 100 | 0 | 100 | 0 | 0 |
| Opt, 3PH | 31 | 43 | 26 | 40 | 5 | 55 | 4 | 19 | 77 | 69 | 0 | 31 | 73 | 1 | 27 | 28 | 0 | 71 | 1 | 37 | 61 | 41 | 52 | 7 |

**Table 4.** Prosumer surplus shifting on the three phases in network R28.

| Prosumer | P3 | P6 | P7 | P10 | P15 | P21 | P25 | P27 |
|---|---|---|---|---|---|---|---|---|
| Initial | B | A | C | A | A | A | B | B |
| Opt, 1PH | C | A | C | C | A | C | B | A |
| Opt, 3PH | ABC | ABC | ABC | AC | ABC | AC | ABC | ABC |

The detailed results regarding the power losses are shown in Table 5 and Figure 10, corresponding to the solutions in Table 3 and the 06:00–18:00 interval, where PV generation is present.

**Table 5.** Power losses in network R28, hourly and total, in the interval 06:00–18:00, kW.

| Scenario | h06 | h07 | h08 | h09 | h10 | h11 | h12 | h13 | h14 | h15 | h16 | h17 | h18 | ΔP Total |
|---|---|---|---|---|---|---|---|---|---|---|---|---|---|---|
| Initial | 0.410 | 0.658 | 1.229 | 0.406 | 0.419 | 0.441 | 0.582 | 0.671 | 0.412 | 0.403 | 0.879 | 0.945 | 1.409 | 8.864 |
| Opt, 1PH | 0.196 | 0.176 | 0.395 | 0.197 | 0.231 | 0.218 | 0.282 | 0.268 | 0.293 | 0.179 | 0.263 | 0.644 | 0.479 | 3.821 |
| Opt, 3PH | 0.155 | 0.175 | 0.352 | 0.093 | 0.109 | 0.090 | 0.145 | 0.126 | 0.112 | 0.090 | 0.138 | 0.469 | 0.345 | 2.398 |

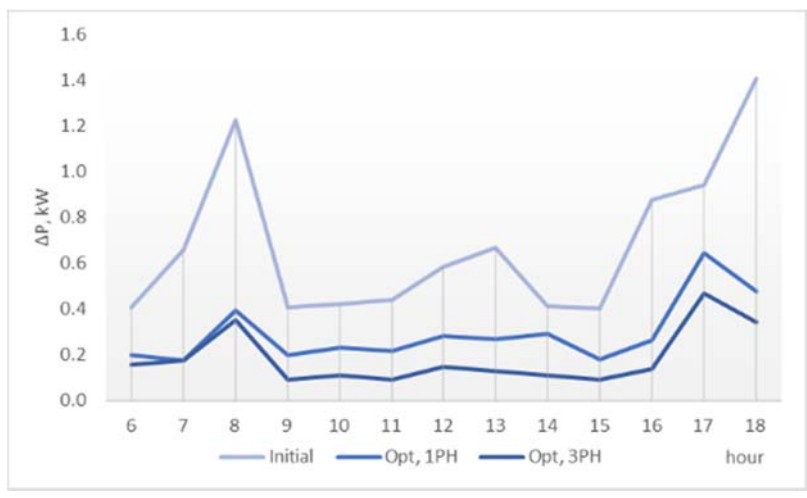

**Figure 10.** Hourly losses in network R28.

The superiority of the proposed method is also visible from the modified phase load profiles of the network, in Figure 11 (one-phase surplus redistribution) and Figure 12 (three-phase surplus distribution), where the balancing effect of shifting the surplus can be compared to the initial operation conditions in Figure 9. The effects are seen only in the 06:00–18:00 interval. The three-phase PSM achieves the best phase load balancing through prosumer surplus shifting.

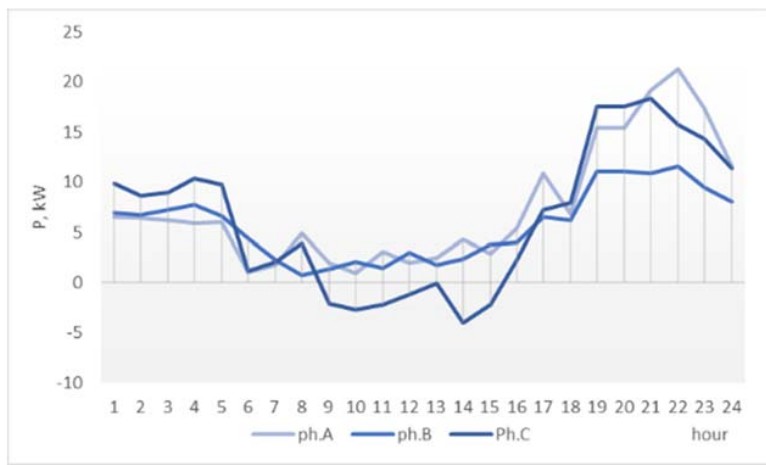

**Figure 11.** Actual bus demand in network R28 (aggregated demand and generation)—one-phase PSM.

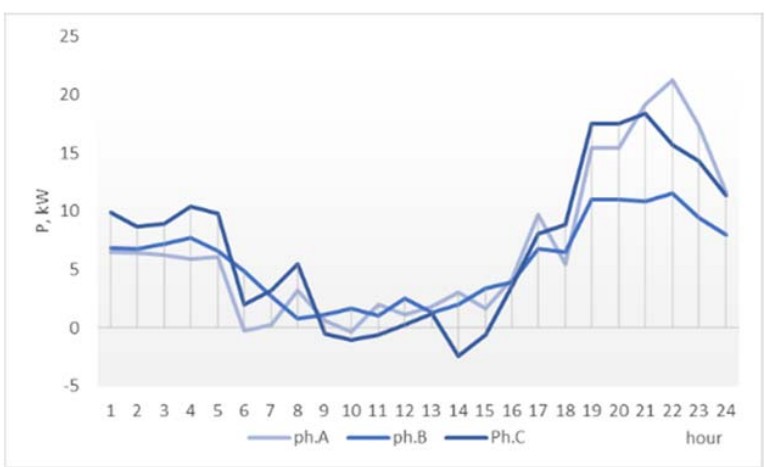

**Figure 12.** Actual bus demand in network R28 (aggregated demand and generation)—three-phase PSM.

*4.2. Results for Network R121*

The other type of network used in the study has lower aggregated consumption (Figure 13), lower prosumer generation (Figure 14), and lower prosumer surplus (Figure 15).

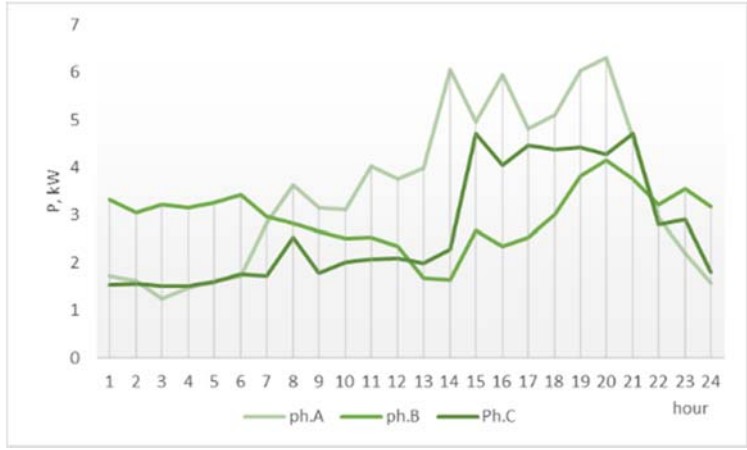

**Figure 13.** Demand in network R121.

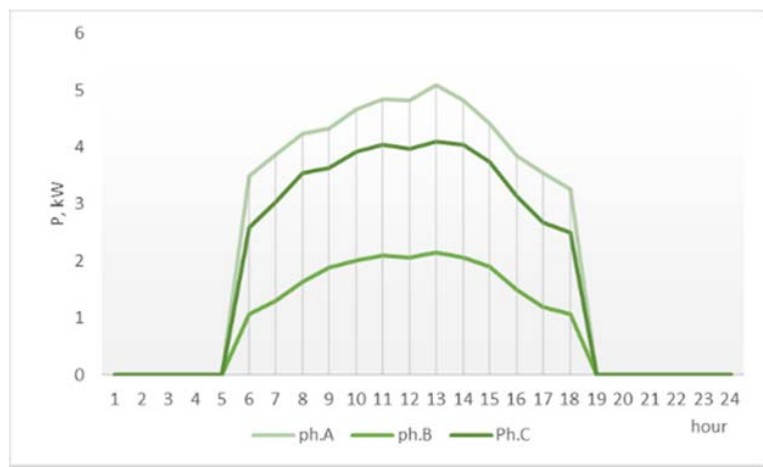

**Figure 14.** Generation in network R121.

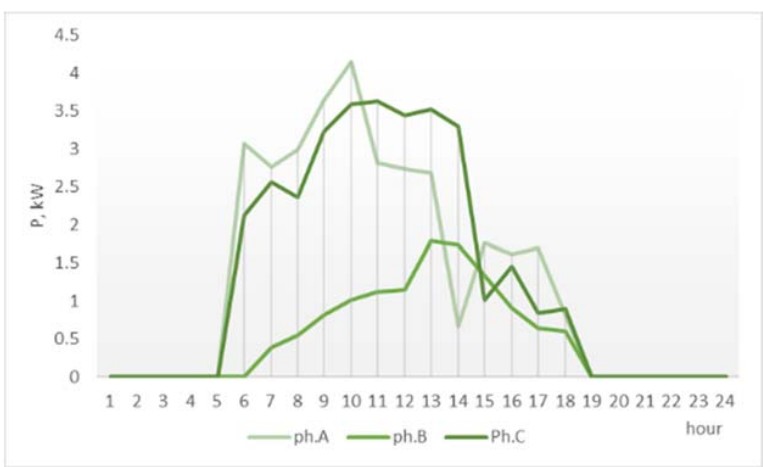

**Figure 15.** Prosumer surplus in network R121.

The prosumer surplus exceeds the local demand (Figure 16), a scenario that results in reversed power flows upwards the MV/LV substation and a lesser effect on reducing the power losses by the PSM algorithm.

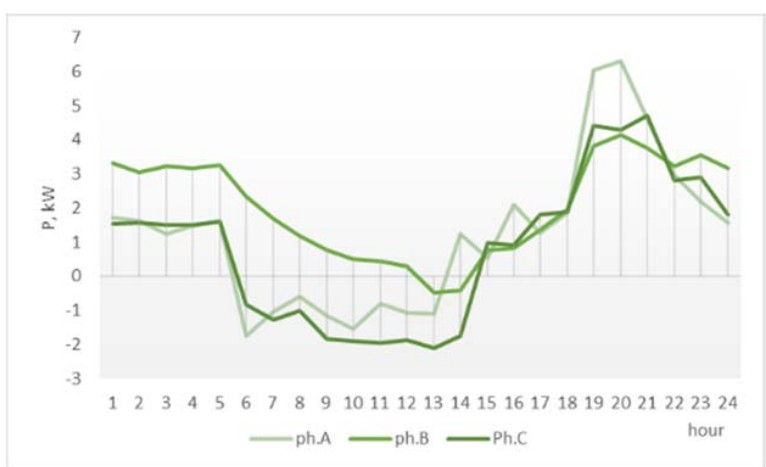

**Figure 16.** Actual bus demand in network R121 (aggregated demand and generation).

The new phase distribution of the prosumer surplus is presented in Tables 6 and 7. Again, both methods of shifting the surplus improved the phase load balancing and

decreased the power losses from 1.86 kW to 1.77 kW in the one-phase optimization and to 1.35 kW in the three-phase optimization. The effect is reduced compared to network R28, because of the complexity of the network and the lower amount of surplus available for balancing. This is particularly visible in Figures 17 and 18, depicting the phase loading obtained after PSM. The detailed results regarding the power losses are presented in Table 8 and Figure 19.

**Table 6.** Surplus redistribution solutions for network R121, in percent.

| Prosumer | P18 | | | P27 | | | P37 | | | P39 | | | P56 | | | P63 | | | P119 | | | P85 | | |
| --- | --- | --- | --- | --- | --- | --- | --- | --- | --- | --- | --- | --- | --- | --- | --- | --- | --- | --- | --- | --- | --- | --- | --- | --- |
| Phase | A | B | C | A | B | C | A | B | C | A | B | C | A | B | C | A | B | C | A | B | C | A | B | C |
| Initial | 0 | 0 | 100 | 100 | 0 | 0 | 0 | 0 | 100 | 0 | 0 | 100 | 100 | 0 | 0 | 0 | 100 | 0 | 0 | 100 | 0 | 100 | 0 | 0 |
| Opt, 1PH | 0 | 0 | 100 | 100 | 0 | 0 | 0 | 100 | 0 | 0 | 0 | 100 | 100 | 0 | 0 | 0 | 0 | 100 | 0 | 100 | 0 | 100 | 0 | 0 |
| Opt, 3PH | 24 | 26 | 50 | 72 | 5 | 23 | 1 | 28 | 72 | 20 | 22 | 58 | 53 | 33 | 14 | 40 | 60 | 1 | 38 | 35 | 27 | 88 | 8 | 4 |

**Table 7.** Prosumer surplus shifting on the three phases in network R121.

| Prosumer | P18 | P27 | P37 | P39 | P56 | P63 | P119 | P85 |
| --- | --- | --- | --- | --- | --- | --- | --- | --- |
| Initial | C | A | C | C | A | B | B | A |
| Opt, 1PH | C | A | B | C | A | C | B | A |
| Opt, 3PH | ABC | ABC | ABC | ABC | ABC | ABC | ABC | ABC |

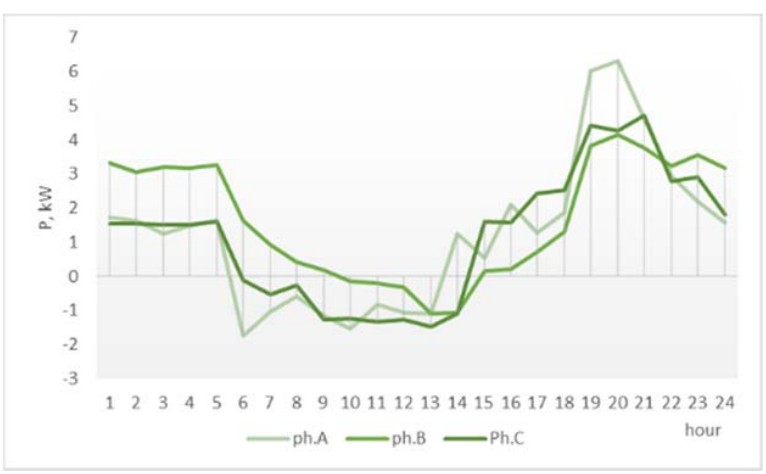

**Figure 17.** Actual bus demand in network R121 (aggregated demand and generation)—one-phase PSM.

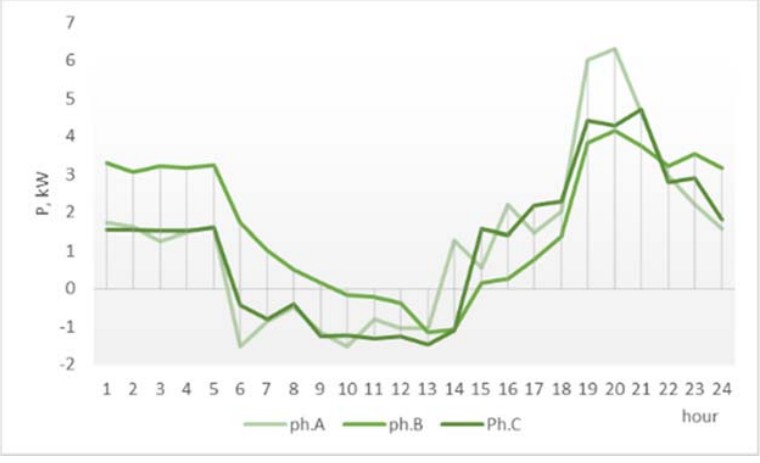

**Figure 18.** Actual bus demand in network R121 (aggregated demand and generation)—three-phase PSM.

**Table 8.** Power losses in network R121, hourly and total, in the interval 06:00–18:00, kW.

| Scenario | h06 | h07 | h08 | h09 | h10 | h11 | h12 | h13 | h14 | h15 | h16 | h17 | h18 | ΔP Total |
|---|---|---|---|---|---|---|---|---|---|---|---|---|---|---|
| Initial | 0.193 | 0.093 | 0.076 | 0.105 | 0.152 | 0.101 | 0.107 | 0.192 | 0.159 | 0.101 | 0.267 | 0.183 | 0.135 | 1.863 |
| Opt, 1PH | 0.177 | 0.072 | 0.059 | 0.102 | 0.149 | 0.063 | 0.065 | 0.116 | 0.119 | 0.156 | 0.314 | 0.222 | 0.160 | 1.775 |
| Opt, 3PH | 0.157 | 0.049 | 0.031 | 0.077 | 0.109 | 0.044 | 0.044 | 0.063 | 0.080 | 0.091 | 0.265 | 0.183 | 0.158 | 1.351 |

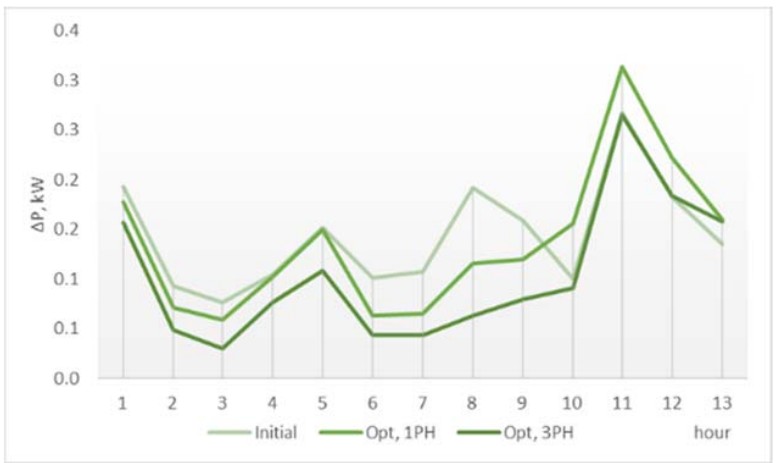

**Figure 19.** Hourly losses in network R121.

As Figures 16–18 show, the initial phase distribution of the prosumers was close to the optimal solution possible with one-phase shifting. By using unbalanced three-phase PSM, the losses were reduced to a significantly lower value, with a 27% drop compared to the initial case.

## 5. Conclusions

The study developed in this paper addresses a problem that is becoming frequent in Romanian residential electricity distribution networks, with the increase in the number of prosumers incentivized by government regulations and subsidy schemes.

The newly built residential suburbs usually have high consumption density confined in a narrow area, with microgrid characteristics. The electricity supply infrastructure is sized accordingly and it is able to accommodate a growing number of prosumers. On the other hand, the older networks were designed with lower consumption requirements and supply larger areas with scattered consumption nodes.

The proliferation of the prosumers must be managed by network operators considering the proximity of both types of supply, over which the new prosumers must be integrated with minimal negative effects on the economic and technical operation of the electricity distribution infrastructure.

The algorithm built by the authors focuses on minimizing the power losses in microgrids with prosumer presence. The study discusses different types of network size, configuration, and consumption patterns. To allow a meaningful comparison regarding the impact of the proposed optimization, the same number of prosumers was considered in different networks, from geographical areas with different PV generation potential.

The results confirmed that by using the capabilities of smart grid, remote control, and the envisioned advances in inverter technology to be deployed in future developments of electricity supply infrastructure, it is possible to turn the presence of intermittent renewable generation into a tool available to the network operator for improving the operation of its supply infrastructure. The proposed algorithm can be scaled for multiple network configurations, load sampling rates, or analysis intervals, according to the needs of local utilities or microgrids.

Future work will consider the implementation of the storage capabilities and the interaction between utility requirements and comfort preferences set by individual consumers using smart home energy management systems.

**Author Contributions:** Conceptualization, M.G., B.-C.N. and O.I.; methodology, O.I.; software, M.G., G.G. and O.I.; validation, B.-C.N.; formal analysis, M.G.; data curation, B.-C.N.; writing—original draft preparation, O.I.; writing—review and editing, B.-C.N. and G.G. All authors have read and agreed to the published version of the manuscript.

**Funding:** This research received no external funding.

**Conflicts of Interest:** The authors declare no conflict of interest.

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
