# Peer review of "A Phase Generation Shifting Algorithm for Prosumer Surplus Management in Microgrids Using Inverter Automated Control"

_electronics, doi:10.3390/electronics10222740_

Round 1

Reviewer 1 Report

  This type of research is necessary in contexts where residential photovoltaic installations are growing, so I do not consider it to be particularly novel. This work shows a methodology to detect the incidence in the network, which has been developed for various contexts, detecting the implications of surpluses. However, it is advisable to do it under extreme scenarios, which can be done on summer days with high irradiation , and if possible by detecting days of minimum consumption, even this second tends to be more complex. However, sizing with high irradiation days in summer is easily possible with simulators, which has not been developed in this study. Images must be improved!

Reviewer 2 Report

Dear authors, 

Some suggestions to improve the quality of the paper: 

Fig. 2 - improve the definition of the picture: it is quite hard to read.

Line 185 - flowchart from Figure 2, the chages being: changes instead of chages

Figures appearance order - I would suggest to present figures at the same order as it is explained in the text. By doing so, fig. 4 and 5 would be inverted.

Line 294 - Chapter 5 and not 4; I would suggest to change the title to “Conclusions”

Author Response

Dear Reviewer,

Reviewer 3 Report

I hope this letter finds you very well, the aim of these few lines is to encourage you to keep up working in this prominent area. The further recommendations were obtained after reading in detail your paper. The recommendations are as follows:

  1.      It is highly required a deeper English grammar since there are plenty of mistakes spreads in the paper.
  2. You are misleading the readers since you are changing the AIM or the OBJECTIVE of your research paper across it. You are requested to define what is your true objective?
  3. The quality of figure must be improved.
  4. I kindly invite you to add a result

Author Response

Dear Reviewer,

Round 2

Reviewer 1 Report

No more comments. The authors have partially solved the inquiries and the no resolved ones, have done the required explaination

Reviewer 3 Report

I am pleased to inform you that your paper "A Phase Generation Shifting Algorithm for Prosumer Surplus Management in Microgrids using Inverter Automated Control" has been accepted for publication in Electronics.